# Role of Monomer/Tetramer Equilibrium of Rod Visual Arrestin in the Interaction with Phosphorylated Rhodopsin

**DOI:** 10.3390/ijms24054963

**Published:** 2023-03-04

**Authors:** Yasushi Imamoto, Keiichi Kojima, Ryo Maeda, Yoshinori Shichida, Toshihiko Oka

**Affiliations:** 1Department of Biophysics, Graduate School of Science, Kyoto University, Kyoto 606-8502, Japan; 2Research Organization for Science and Technology, Ritsumeikan University, Kusatsu 525-8577, Japan; 3Department of Physics, Faculty of Science, Shizuoka University, Shizuoka 422-8529, Japan; 4Nanomaterials Research Division, Research Institute of Electronics, Shizuoka University, Shizuoka 422-8529, Japan

**Keywords:** G protein coupled receptor, desensitization, cell signaling, tetramerization, stoichiometry, small angle X-ray scattering, wide angle X-ray scattering

## Abstract

The phototransduction cascade in vertebrate rod visual cells is initiated by the photoactivation of rhodopsin, which enables the activation of the visual G protein transducin. It is terminated by the phosphorylation of rhodopsin, followed by the binding of arrestin. Here we measured the solution X-ray scattering of nanodiscs containing rhodopsin in the presence of rod arrestin to directly observe the formation of the rhodopsin/arrestin complex. Although arrestin self-associates to form a tetramer at physiological concentrations, it was found that arrestin binds to phosphorylated and photoactivated rhodopsin at 1:1 stoichiometry. In contrast, no complex formation was observed for unphosphorylated rhodopsin upon photoactivation, even at physiological arrestin concentrations, suggesting that the constitutive activity of rod arrestin is sufficiently low. UV-visible spectroscopy demonstrated that the rate of the formation of the rhodopsin/arrestin complex well correlates with the concentration of arrestin monomer rather than the tetramer. These findings indicate that arrestin monomer, whose concentration is almost constant due to the equilibrium with the tetramer, binds to phosphorylated rhodopsin. The arrestin tetramer would act as a reservoir of monomer to compensate for the large changes in arrestin concentration in rod cells caused by intense light or adaptation.

## 1. Introduction

G-protein-coupled receptors (GPCRs) compose the largest family of receptor proteins located on the cell surface. It is known that they have surprisingly diverse physiologic functions while sharing the characteristic structural motif composed of seven transmembrane helices [1]. Recent progress in structural biology has provided high-resolution structures of various GPCRs in resting and ligand-activated states. As a consequence, it is accepted that the common activation mechanism of GPCRs is the outward movement of transmembrane helices, especially TM6, to generate the cavity to bind to the Gα subunits of cognate G proteins [1,2], which activate effector molecules. The stream of signal transduction initiated by GPCRs is quenched by the phosphorylation of the receptors, followed by the binding of arrestin [3]. In addition to the desensitization, arrestin itself initiates the signaling cascade. This framework of signal transduction is also shared by most GPCR systems.

The GPCR family, which responds to light, is called the opsin family [4]. Visual arrestin is the most abundant regulatory protein in visual transduction. It binds to photoactivated and phosphorylated rhodopsin to block the activation of visual G-protein transducin (G_t_), but the signaling pathway initiated by arrestin has not been identified in the transduction system in the visual cells. Humans have several arrestin genes. Among them, rod arrestin shows the highest selectivity for GPCRs. While cone arrestin binds to phosphorylated cone pigments the most effectively, it moderately binds to unphosphorylated cone pigments or non-visual GPCRs. β-arrestin binds to phosphorylated rhodopsin, cone pigments, and non-visual GPCRs, but little binds to unphosphorylated ones [5]. Meanwhile, arrestins show the difference in the manner of oligomerization. Rod arrestin self-associates to form a tetramer at physiologic concentrations [6,7,8], cone arrestin does not oligomerize, and β-arrestin forms a linear oligomer [9,10]. These findings strongly suggest that the diversity of arrestin’s nature is closely correlated with its physiologic functions.

Recent crystallography has demonstrated the architecture of the complex of rhodopsin and arrestin [11]. The finger loop of arrestin is inserted into the cytoplasmic cavity of metarhodopsin II (Meta-II), suggesting the competitive block of G_t_ activation. This high-resolution snapshot is highly informative, but the dynamics of the interactions between proteins should be analyzed to understand the signal transduction process in which multiple proteins are involved. The interaction between phosphorylated Meta-II and arrestin in the presence of tetrameric arrestin is especially important because a significant portion of rod arrestin forms tetramers at physiological concentrations (2–8 mg/mL [12,13]).

Nowadays, a lot of techniques to analyze the interactions between soluble proteins are available. However, those for membrane proteins are still limited, while most receptor proteins are membrane-embedded. Most biochemical or structural analyses of protein complexes involving membrane proteins have been carried out in the detergent system. However, because detergent causes various artifacts, it is desirable to study in a membrane environment. However, native membranes or liposomes are not suitable for biophysical analysis due to their turbidity and heterogeneity of particle size. To overcome these flaws, nanodiscs are one of the most promising solutions [14]. One nanodisc is composed of one or two GPCRs, two membrane scaffold proteins, and a lipid bilayer, and the diameter is around 11 nm.

We have developed the solution X-ray scattering experiments [15] of the proteins involved in the visual transduction cascade. The scattering profile of the solution X-ray scattering includes the distance information corresponding to the scattering vector (*Q*). Therefore, by changing the scattering angle of interest, which is readily achieved by changing the sample-detector distance, one can obtain the small-angle X-ray scattering (SAXS) to detect the protein-protein interaction [8] as well as the wide-angle X-ray scattering (WAXS) to detect the intramolecular rearrangement of the secondary structure elements [16,17] using a similar experimental setup and sample. In the present work, we conducted the WAXS and SAXS measurements using nanodiscs to detect the photoactivation of rhodopsin, followed by the association with rod arrestin, which is the key event in the shut-off mechanism of visual transduction. 

We demonstrated that arrestin is bound to phosphorylated Meta-II in 1:1 stoichiometry even in the presence of a significant amount of tetrameric arrestin at physiological concentrations. In addition, the UV-visible spectroscopy demonstrated that the rate of formation of the rhodopsin/arrestin complex was in good agreement with the concentration of arrestin monomer. Because the concentration of monomeric arrestin is almost constant at physiological concentration due to the concentration-dependent equilibrium between the monomer and tetramer, phosphorylated Meta-II is deactivated by binding to monomeric arrestin to keep the quenching kinetics stationary. 

## 2. Results

### 2.1. Light-Induced Structural Change of Phosphorylated Rhodopsin

Rhodopsin is photoconverted to the G_t_-activating intermediate metarhodopsin II (Meta-II). In Meta-II, the transmembrane helices are significantly rearranged, and the generated cytoplasmic cavity interacts with the C-terminus of the G_t_α subunit [17,18,19]. However, phosphorylation of rhodopsin partially suppresses G_t_ activation [20]. Thus, we assessed whether phosphorylation of rhodopsin affects the light-induced helical rearrangement by wide-angle X-ray scattering (WAXS) [16,17,21].

In the range of 0.1 < *Q* < 1.0 Å^−1^ (~6 < *D* < ~60 Å), the scattering intensity is sensitive to interference between GPCR transmembrane helices. Therefore, rearrangement of helices can be detected by measuring the intensity change in this *Q* region [16,17,21]. The WAXS measurements were carried out using nanodiscs containing native rhodopsin, bleached/regenerated rhodopsin, and bleached/phosphorylated/regenerated rhodopsin (nRh/ND, rRh/ND, and P-rRh/ND, respectively) before and after photoactivation. Figure 1 shows the difference between the WAXS curves that were calculated. The characteristic peak at 0.2 Å^−1^ and valley at 0.6 Å^−1^ in Meta-II are derived from the outward movement of TM6 and the displacement of the cytoplasmic sides of TM3 and TM5, respectively [17]. The valley at 0.6 Å^−1^ is notable because displacement of the cytoplasmic sides of TM3 and TM5 stabilizes the active conformation of rhodopsin [17]. These scattering intensity changes derived from helical rearrangements are comparable, indicating that the structure of Meta-II is not altered by the phosphorylation, as previously reported by FTIR analysis [22]. The electrostatic effect of phosphates is likely to suppress G_t_ activation.

### 2.2. Small-Angle X-ray Scattering of Nanodiscs

Nanodiscs are phospholipid bilayer fragments surrounded by two membrane scaffold protein (MSP) molecules. The size of nanodiscs (diameter around 11 nm) is determined by the length of MSP molecules, and one or two GPCR molecules are incorporated into one nanodisc. In this study, nanodiscs containing one rhodopsin molecule were prepared because visual arrestin binds to monomeric rhodopsin [25]. The SAXS profiles of bare nanodiscs and nRh/ND were measured (Figure 2), and they were consistent with the previous reports [26,27]. The scattering profiles at 12–49 µM agreed with each other (Figure 2b), indicating that no aggregation of nanodiscs occurred at this concentration.

The SAXS profiles were quantitatively analyzed by Guinier analysis, which gives the forward scattering (*I*(0)) and the radius of gyration (*R*_g_) (Equation (16)). The Guinier plot of nRh/ND (Figure 2c) shows the linear region, and the slope of the linear region is constant at all concentrations, indicating that nRh/ND is monodispersed. *R*_g_ of nRh/ND in our preparation was estimated to be 64.4 Å (Figure 2f), which is greater than that of previous reports (52 Å for rhodopsin nanodiscs and 63 Å for bare nanodiscs [28]). *I*(0) is proportional to the square of the molecular weight of the scatterer (Equation (17)) if the electron density of the particle is uniform. However, because the electron density of the lipid bilayer is lower than that of proteins, we tested whether *I*(0) could be a measure of the molecular weight of the nanodiscs.

Ovalbumin (43 K) was used as the molecular weight standard (Figure 2d). *I*(0) for ovalbumin was estimated from the Guinier plot of the scattering profile of ovalbumin at 23–140 µM. Because ovalbumin is monodispersed in this range of concentrations [8], *I*(0) is proportional to the molar concentration (Figure 2e). Then, *I*(0) for nRh/ND was estimated from the Guinier plot by linear extrapolation to *Q*^2^ = 0 (Figure 2c), and *I*(0) for nRh/ND was plotted against the molar concentration of nRh/ND (Figure 2e). Assuming the inter-particle interaction of the X-ray is negligible, the slope of the *I*(0) vs molar concentration is proportional to the square of the molecular weight. In fact, the plot of nRh/ND was linear (Figure 2e). The apparent molecular weight of nRh/ND estimated from the slopes of nRh/ND and ovalbumin was 111 K, which is in good agreement with the sum of the molecular weights of one rhodopsin molecule and two MSP molecules (108 K). Therefore, *I*(0) can be practically used as a measure of the molecular weight of nanodiscs.

### 2.3. Formation of the Complex of Photoactivated Rhodopsin and Arrestin

The SAXS profiles of the mixture of arrestin (0.25–4 mg/mL) and rRh/ND or P-rRh/ND were recorded before and after photoexcitation. *I*(0) of the sample was then estimated by the linear extrapolation of the Guinier plot to *Q*^2^ = 0 (Figure 3).

*I*(0) of the mixture of rRh/ND and arrestin was not changed by irradiation at any arrestin concentration (Figure 3a,b), indicating that no complex is formed without the phosphorylation of rhodopsin. In contrast, *I*(0) of the mixture of P-rRh and arrestin was significantly increased by the irradiation (Figure 3c,d), indicating the formation of a complex. *I*(0) before and after irradiation was divided by the total weight concentration of proteins (arrestin, MSP, and rhodopsin) (*I*(0)/*C*_W_) and plotted against the total arrestin concentration (Figure 4).

To estimate *I*(0) in the absence of the interaction between arrestin and rRh or P-rRh, the scattering profiles of arrestin, rRh/ND, and P-rRh/ND were separately measured, and *I*(0) was estimated by the Guinier plot. The sum of *I*(0) of arrestin and rRh or P-rRh/ND was divided by *C*_W_ and plotted against arrestin concentration (Figure 4, blue).

A significant increase in *I*(0)/*C*_W_ of P-rRh was observed at low arrestin concentrations. The increase in *I*(0) is caused by the formation of the arrestin/P-rRh complex, whereas the decrease in *I*(0) is caused by the dissociation of the arrestin tetramer. At low concentrations (<1 mg/mL), arrestin is mainly in monomer (45 K), so *I*(0) increases as arrestin concentration increases. In contrast, at high arrestin concentrations, a significant portion of excess arrestin is in tetramer (180 K). The molecular weight of the complex of P-rRh/ND and arrestin monomer (108 + 45 K) is comparable to that of arrestin tetramer (180 K), whereas that of the complex of P-rRh/ND and arrestin tetramer (108 + 180 K) is significantly greater. Therefore, small increases in *I*(0) at high arrestin concentrations strongly suggest that *I*(0) increases by the formation of a 1:1 complex of P-rRh/ND and arrestin is canceled by arrestin tetramer dissociation. To examine that, changes in *I*(0)/*C*_W_ were quantitatively analyzed based on the model in which the arrestin tetramer and the1:1 arrestin/rhodopsin complex are formed (Figure 5).

At the current experimental concentration, arrestin forms a tetramer in a concentration-dependent manner. Because the cooperativity of self-association of arrestin is high, the formation of the tetramer is virtually expressed by the equilibrium between monomer and tetramer [8] (Figure 5). For simplification, the tetramerization of arrestin is expressed as follows:(1)$$\frac{{\left[\mathrm{Arr}\right]}^{4}}{\left[{\mathrm{Arr}}_{4}\right]}=K_{\mathrm{Arr}}^{3}$$

The concentration dependence of *I*(0)/*C*_W_ of arrestin was measured, and $K_{\mathrm{Arr}}$ was estimated to be 41.5 µM (Appendix A and Table 1).

P-rRh and arrestin combine to form the complex with a dissociation constant of $K_{\mathrm{complex}}$ (Figure 5):(2)$$\frac{\left[P\text{-}\mathrm{rRh}\right]\left[\mathrm{Arr}\right]}{\left[P\text{-}\mathrm{rRh}\cdot \mathrm{Arr}\right]}=K_{\mathrm{complex}}$$

On the other hand, the total amount of P-rRh is constant and equal to the sum of free P-rRh/ND and the complex of P-rRh/ND and arrestin.
(3)$${\left[P\text{-}\mathrm{rRh}\right]}_{\mathrm{total}}=\left[P\text{-}\mathrm{rRh}\right]+\left[P\text{-}\mathrm{rRh}\cdot \mathrm{Arr}\right]$$

Similarly, the total amount of arrestin is constant and equal to the sum of the free arrestin monomer, the arrestin tetramer (×4), and the complex of P-Rh/ND and arrestin.
(4)$${\left[\mathrm{Arr}\right]}_{\mathrm{total}}=\left[\mathrm{Arr}\right]+4\times \left[{\mathrm{Arr}}_{4}\right]+\left[P\text{-}\mathrm{rRh}\cdot \mathrm{Arr}\right]$$

Using Equations (1)–(3), the concentration of each component is expressed using the concentration of [Arr] (free arrestin monomer) as follows:(5)$$\left[P\text{-}\mathrm{rRh}\right]=\frac{K_{\mathrm{complex}}{\left[P\text{-}\mathrm{rRh}\right]}_{\mathrm{total}}}{K_{\mathrm{complex}}+\left[\mathrm{Arr}\right]}\;\left[P\text{-}\mathrm{rRh}\cdot \mathrm{Arr}\right]=\frac{\left[\mathrm{Arr}\right]{\left[P\text{-}\mathrm{rRh}\right]}_{\mathrm{total}}}{K_{\mathrm{complex}}+\left[\mathrm{Arr}\right]}$$

Equations (1), (4) and (5) imply that when [Arr] is given, ${\left[\mathrm{Arr}\right]}_{\mathrm{total}}$, $\left[{\mathrm{Arr}}_{4}\right]$, [P-rRh], and [P-rRh·Arr] are calculated using $K_{\mathrm{Arr}}$ and $K_{\mathrm{complex}}$, which further give *I*(0) of the mixture using the molecular weights of arrestin (45 K), rhodopsin (42 K), and MSP (33 K) (Equation (17)) as follows:(6)$$I\left(0\right)\propto 45,000^{2}\times \left[\mathrm{Arr}\right]+{\left(45,000\times 4\right)}^{2}\times \left[{\mathrm{Arr}}_{4}\right]+{\left(42,000+33,000\times 2\right)}^{2}\times \left[P\text{-}\mathrm{rRh}\right]\;+{\left(42,000+33,000\times 2+45,000\right)}^{2}\times \left[P\text{-}\mathrm{rRh}\cdot \mathrm{Arr}\right]$$

${\left[\mathrm{Arr}\right]}_{\mathrm{total}}$, $\left[{\mathrm{Arr}}_{4}\right]$, [P-rRh], [P-rRh·Arr] and I(0) were calculated by varying [Arr]. Then *I*(0)/*C*_W_ was plotted against the total arrestin concentration. Based on the least mean squared deviation between the calculated curve and experimental data, $K_{\mathrm{complex}}$ was determined for the dark state and photoactivated state (Figure 5). The same analysis was carried out for rRh, and $K_{\mathrm{complex}}$ is summarized in Table 1. It should be noted that this analysis includes both monomer and tetramer equilibrium and complex formation. 

*I*(0)/*C*_W_ for rRh was slightly greater than that without interaction (Figure 4a), but $K_{\mathrm{complex}}$ was not altered by photoactivation, suggesting that a small amount of arrestin non-specifically binds to nanodiscs. $K_{\mathrm{complex}}$ for P-rRh in the dark shows the binding of arrestin and the phosphorylated C-terminus of P-rRh (Figure 5). Because rhodopsin is phosphorylated after photoactivation in rod cells, this interaction would be an artifact but mimics the pre-coupling complex, in which interaction between the finger loop and cytoplasmic cavity is absent (see Section 3.1). *I*(0)/*C*_W_ for photoactivated P-rRh was well reproduced by a 1:1 complex model at all concentrations (Figure 4b), indicating that arrestin bound to rhodopsin does not undergo concentration-dependent self-association like free arrestin.

### 2.4. Formation of Extra-Meta-II by Binding of Arrestin, Detected by UV-Visible Spectroscopy

The SAXS experiment demonstrated that P-rRh and arrestin form a 1:1 complex, strongly suggesting that the free monomer of arrestin binds to rhodopsin. Because the concentration of arrestin monomer is not largely varied as compared to tetramer [8], the rate of arrestin binding is likely to be nearly equal regardless of total arrestin concentration if arrestin monomer specifically binds to P-rRh. Thus, the formation of extra-Meta-II was measured at various arrestin concentrations by time-resolved UV-visible spectroscopy (Figure 6).

P-rRh is converted to Meta-I shortly after flash excitation (< 1 ms) and then converted to Meta-II to form the equilibrium between Meta-I and Meta-II. In the absence of arrestin, this equilibrium is formed at 100 ms and stable up to 10 s (Figure 6a). While the kinetics of the formation of Meta-II are complicated due to the heterogeneity of intermediates and branched reactions [29], the absorbance change at 480 nm (the absorption maximum of Meta-I) plotted against the time after excitation was expressed using a double-exponential function (Figure 6c, top).
(7)$$\Delta A_{480}=A_{0}+A_{1}e^{-k_{1}t}+A_{2}e^{-k_{2}t}$$

On the other hand, in the presence of arrestin, the transient difference spectra at 1 s and 10 s after excitation disagreed with those at 100 ms, indicating that Meta-I and Meta-II equilibrium was further biased toward Meta-II due to arrestin binding (Figure 6b). The absorbance changes at 480 nm in the presence of 0.5–8 mg/mL arrestin were fitted with a triple-exponential function, to which the term of formation of extra-Meta-II was added.
(8)$$\Delta A_{480}=A_{0}+A_{1}e^{-k_{1}t}+A_{2}e^{-k_{2}t}+\Delta \Delta Ae^{-k_{\mathrm{app}}t}$$
where ΔΔA is the absorbance decrease at 480 nm due to the formation of extra-Meta-II, and $k_{\mathrm{app}}$ is the apparent rate constant for the formation of extra-Meta-II. The absorbance changes at 480 nm were global-fitted with Equation (7) (0 mg/mL arrestin) or Equation (8) (0.5–8 mg/mL arrestin), in which $A_{1}$, $k_{1}$, $A_{2}$, and $k_{2}$ were set as global variables (Figure 6c). 

It is assumed that either the monomer or the tetramer of arrestin (${\mathrm{Arr}}_{x}$) binds to P-rRh as follows: (9)$${\mathrm{Arr}}_{x}+P\text{-}\mathrm{rRh}\overset{k_{a}}{\underset{k_{d}}{⇄}}{\mathrm{Arr}}_{x}\cdot P\text{-}\mathrm{rRh}$$

Hence,
(10)$$\frac{d\left[{\mathrm{Arr}}_{x}\cdot P\text{-}\mathrm{rRh}\right]}{dt}=k_{a}\left[{\mathrm{Arr}}_{x}\right]\left[P\text{-}\mathrm{rRh}\right]-k_{d}\left[{\mathrm{Arr}}_{x}\cdot P\text{-}\mathrm{rRh}\right]$$


Because arrestin is present in excess over rhodopsin in these measurements, the concentration of arrestin is approximated by a constant ($k_{a}\left[{\mathrm{Arr}}_{x}\right]=k_{a}^{\prime }$) (pseudo first order reaction). $\left[{\mathrm{Arr}}_{x}\cdot P\text{-}\mathrm{rRh}\right]$ at time *t* is expressed as follows:(11)$${\left[{\mathrm{Arr}}_{x}\cdot P\text{-}\mathrm{rRh}\right]}_{t}={\left[P\text{-}\mathrm{rRh}\right]}_{0}\frac{k_{a}^{\prime }}{k_{a}^{\prime }+k_{d}}\left(1-e^{-\left(k_{a}^{\prime }+k_{d}\right)t}\right)$$

In this equation, $k_{\mathrm{app}}$ is equal to the sum of $k_{a}^{\prime }+k_{d}$.
(12)$$k_{\mathrm{app}}=k_{a}^{\prime }+k_{d}$$

The fraction of ${\mathrm{Arr}}_{x}\cdot P\text{-}\mathrm{rRh}$ (F) is expressed by $k_{a}^{\prime }$ and $k_{d}$.
(13)$$F=\frac{k_{a}^{\prime }}{k_{a}^{\prime }+k_{d}}$$

ΔΔA is proportional to F at each arrestin concentration. Here, *F* was calculated assuming that $\left[{\mathrm{Arr}}_{x}\cdot P\text{-}\mathrm{rRh}\right]$ is fully formed at 8 mg/mL. From Equations (12) and (13),
(14)$$k_{\mathrm{app}}F=k_{a}^{\prime }=k_{a}\left[{\mathrm{Arr}}_{x}\right]$$

$k_{\mathrm{app}}F$ was plotted against the total arrestin concentration in the reaction mixture (Figure 6d). It was in good agreement with monomer concentration, indicating that P-rRh specifically binds to arrestin monomer and the tetramer serves as the reservoir of monomer.

## 3. Discussion

### 3.1. Interaction between Rhodopsin and Arrestin

Arrestin is an abundant regulatory protein in the GPCR system. It binds to phosphorylated GPCRs and blocks the activation of G proteins by GPCRs. Rod arrestin expressed in rod visual cells undergoes concentration-dependent tetramerization at physiological concentrations. As a consequence, the concentration of arrestin monomer is almost constant, whereas that of arrestin tetramer is nearly proportional to the total arrestin concentration in the cell. Therefore, whether phosphorylated rhodopsin binds to monomeric or tetrameric arrestin is directly correlated with the kinetics of quenching rhodopsin by arrestin. 

Here, we directly analyzed the complex of arrestin and membrane-embedded rhodopsin by SAXS at the physiological arrestin concentration. While a significant portion of arrestin forms a tetramer, arrestin and P-rRh form a 1:1 complex. It indicates that binding to phosphorylated rhodopsin inhibits the oligomerization of arrestin, whereby arrestin would not be wasted. UV-visible spectroscopy confirmed that the kinetics of extra-Meta-II formation due to arrestin binding were well correlated with arrestin monomer concentration (Figure 6). 

It is reported that arrestin is translocated between the inner and outer segments upon adaptation [30,31,32,33]. The arrestin concentration in light-adapted rod outer segments is much higher than that in dark-adapted ones. Therefore, the concentration of arrestin monomer is buffered by the tetramer, and the efficiency of shutoff by arrestin binding would be maintained at a constant level despite the adaptation state of the rod cell. Equilibrium between monomer and tetramer would also prevent the depletion of arrestin when a large amount of rhodopsin is bleached by intense light.

The structure of the GPCR/arrestin complex has been studied by X-ray crystallography [11,34,35] and single-particle analysis using an electron microscope [36]. These studies demonstrated that the finger loop (β5–β6 loop) located at the middle of the arrestin molecule is inserted into the transmembrane region of GPCR, which is opened by ligand binding. This is consistent with the notion that the amino acid sequence of the arrestin finger loop is similar to the C-terminal tail of the G protein α-subunit, and that the oligopeptide derived from the amino acid sequence of the finger loop (Tyr67-Leu77) acts as a “high affinity peptide” that stabilizes Meta-II [37]. 

Arrestin binds to rhodopsin in multiple steps [38] (Figure 7). The phosphorylated C-terminus of rhodopsin first binds to arrestin (pre-couple). Then the finger loop of arrestin is released, and it binds to the cytoplasmic cavity of photoactivated rhodopsin (full couple) [35]. A good correlation between the rate of extra-Meta-II formation and the concentration of arrestin monomer (Figure 6d) strongly suggests that the pre-coupling determines the rate of arrestin binding. 

Because rhodopsin is phosphorylated after photoactivation in rod cells, the interaction between P-rRh/ND and arrestin in the dark (Figure 5) would be an artifact. However, it would mimic the pre-coupling complex, in which the interaction between the finger loop and cytoplasmic cavity is absent. Arrestin does not bind to photoactivated rRh even at high physiological concentrations (Figure 4a). The finger loop of arrestin would never be released by the induced-fit mechanism to photoactivated rhodopsin, implying that arrestin is not activated without the pre-coupling step. Therefore, due to the pre-coupling mechanisms, the constitutive activity of rod arrestin would be suppressed, and it does not inhibit G_t_ binding to unphosphorylated rhodopsin. Once photoactivated rhodopsin is phosphorylated, its C-terminus activates arrestin by releasing the finger loop, and the high-affinity complex blocks G_t_ binding. The pre-coupling step is essential to switch the activating stage to the quenching stage by unlocking the arrestin activity to bind it to photoactivated rhodopsin.

### 3.2. Solution X-ray Scattering Experiments for the Analysis of Signal Transduction 

So far, the protein/protein interactions in the physiological condition have been difficult to detect. We have developed SAXS using detergent-solubilized rhodopsin to detect the complex formation [8], but the detergent causes the aggregation of proteins. Here we showed that the nanodisc sample is a suitable sample to detect the formation of protein complexes by SAXS measurement. We quantitatively determined the stoichiometry and dissociation constants using *I*(0). On the other hand, WAXS measurement is a powerful tool to detect the helical rearrangement of GPCRs. The current finding that phosphorylation does not alter the structure of Meta-II in the lipid bilayer is consistent with the high-resolution structures of arrestin-bound Meta-II and G_t_-bound Meta-II, which are comparable [11,19].

## 4. Materials and Methods

### 4.1. Sample Preparation

Rod outer segments (ROS) were prepared from fresh bovine retinas as reported previously [8]. ROS were illuminated with room light in the presence or absence of 5 mM ATP overnight. They were supplemented with 11-cis-retinal and kept in the dark overnight to regenerate pigment. Unbleached (native) rhodopsin, bleached/phosphorylated/regenerated rhodopsin, and bleached/regenerated rhodopsin are referred to as nRh, P-rRh, and rRh, respectively.

nRh, P-rRh, or rRh was incorporated into the nanodiscs composed of membrane scaffold protein (MSP1E3D1) and 1-palmitoyl-2-oleoyl-*sn*-glycero-3-phosphocholine (POPC), as reported previously [14,39]. Briefly, rhodopsin was solubilized with n-octyl-β-D-glucoside (OG), which was then mixed with a 10-fold molar excess of MSP1E3D1 and a 750-fold molar excess of POPC. The mixture was dialyzed against a buffer to remove the detergent. Formed nanodiscs containing rhodopsin were isolated by Superdex 200 column chromatography and concanavalin A affinity column chromatography. The molar ratio of rhodopsin and MSP1E3D1 was estimated to be 1:2 (i.e., each nanodisc contains one rhodopsin molecule) by the absorbance ratio at 500 and 278 nm.

Arrestin was isolated from the soluble fraction of the homogenate of fresh bovine retina by the combination of DEAE-cellulose column chromatography and heparin Sepharose column chromatography [40]. The soluble fraction dialyzed against the buffer (10 mM HEPES, 15 mM NaCl, pH 7.0) was loaded on the DEAE-cellulose column, and the eluate from 55 mM NaCl containing arrestin was directly loaded on the heparin Sepharose column [41]. Arrestin was eluted by 0–8 mM phytic acid in the buffer. After 10-fold dilution by the buffer, the sample was loaded on a heparin-Sepharose column and eluted by 150–800 mM NaCl. Arrestin was detected by SDS poly-acrylamide gel electrophoresis. The sample was repeatedly concentrated by ultrafiltration (YM30) and diluted by the buffer (10 mM HEPES, 100 mM NaCl, pH 7.5) to exchange the buffer. Finally, arrestin was concentrated to 11–12 mg/mL.

Ovalbumin, a molecular weight standard (43 K), was purchased from Sigma (St. Louis, MO 68178, USA). It was solubilized by the buffer (10 mM HEPES, 100 mM NaCl, pH 7.5) and centrifuged before the SAXS measurements.

### 4.2. Analysis of Phosphorylation Levels

Phosphorylated rhodopsin was partially digested by the endoprotease Asp-N, which releases the C-terminal 19 amino acid peptide of rhodopsin [42]. The sample was centrifuged, and the supernatant was subjected to matrix-associated laser desorption ionization (MALDI) time-of-flight mass spectroscopy at the Mass Spectrometry Room in the Graduate School of Engineering, Kyoto University, Kyoto, Japan (Appendix A). 2,5-Dihydroxybenzoic acid was used as a matrix.

The ionization efficiency is reduced by phosphorylation [43]. Assuming that the MALDI ionization efficiencies are dependent on phosphorylation level in the same way that electron spray ionization (ESI) is, the mean phosphate to rhodopsin ratio was estimated to be 3.7:1.

### 4.3. Solution X-ray Scattering

Light-induced structural change of rhodopsin was analyzed by wide-angle X-ray scattering (WAXS) measurements at beamline BL40B2 at SPring-8 (Sayo, Japan) [16,17]. For WAXS measurements, RAXIS VII (Rigaku, Tokyo, Japan) was used as a detector, the sample-detector distance was 540 mm, and the wavelength of the X-ray (λ) was 1.00 Å.

Small-angle X-ray scattering (SAXS) measurements were carried out at the Shizuoka University Research Institute of Green Science and Technology, or at beamline BL40B2 at Spring-8. Pilatus 100 K (Dectris, Baden-Daettwil, Switzerland) was used as a detector, the sample-detector distance was 800 mm (Shizuoka University) or 1200 mm (Spring-8), and the wavelength of the X-ray (λ) was 1.54 (Shizuoka University) or 1.25 Å (Spring-8). 

For both WAXS and SAXS measurements, the sample was put in a 2 mm ϕ quartz capillary (Hilgenberg, Malsfeld, Germany) and perpendicularly illuminated with a 150 W fiber light source to which a yellow glass filter was attached (Edmund Optics, Barrington, NJ 08007, USA). Two-dimensional images of scattered X-rays were circularly averaged, and the scattering intensity at *Q* (=4πsinθ/λ) was calculated. 

### 4.4. SAXS Data Analysis

The two-dimensional image obtained by Pilatus was converted to the scattering curve by a circular average. To calculate the scattering profile derived from the protein ($I_{\mathrm{protein}}\left(Q\right)$), the scattering profile for the buffer ($I_{\mathrm{buffer}}\left(Q\right)$) was subtracted from the protein solution ($I_{\mathrm{solution}}\left(Q\right)$) as follows:(15)$$I_{\mathrm{protein}}\left(Q\right)=I_{\mathrm{solution}}\left(Q\right)-I_{\mathrm{buffer}}\left(Q\right)\times \left(1-\bar{\nu }\times C_{w}\right)$$
where $\bar{\nu }$ is the partial specific volume of protein (=0.73 mL/g), and *C*_W_ is the weight concentration of the protein. In the small-angle region (*R*_g_*Q* < 1.3), the scattering intensity is approximated as follows [44]:(16)$$\ln I\left(Q\right)=\ln I\left(0\right)-\frac{R_{g}{}^{2}}{3}Q^{2}$$
where *Q* is the momentum transfer, *I*(*Q*) is the scattering intensity at *Q*, *I*(0) is the forward scattering, and *R*_g_ is the radius of gyration. Thus, the plot of ln *I*(*Q*) against *Q*^2^ (Guinier plot) in the small-angle region was linearly extrapolated to *Q*^2^ = 0, and ln *I*(0) and *R*_g_^2^ were estimated by the Y-intersection and the slope of the regression line, respectively.

*I*(0) per particle is proportional to the square of the total number of electrons in the particle, which is approximately proportional to the molecular weight.
(17)$$\frac{I\left(0\right)}{C_{\mathrm{mol}}}\propto N^{2}\propto {\mathrm{MW}}^{2}$$
where $C_{\mathrm{mol}}$ is the molar concentration of the particle, *N* is the total number of electrons, and MW is the molecular weight of the particle. Thus, *I*(0) divided by weight concentration ($C_{W}=\mathrm{MW}\times C_{\mathrm{mol}}$) is proportional to the molecular weight.
(18)$$\frac{I\left(0\right)}{C_{w}}\propto \mathrm{MW}$$

Because the sample for SAXS was a mixture of nanodiscs and arrestin, *I*(0) was divided by the weight concentration of the total proteins, where *I*(0)/*C*_W_ is proportional to the average molecular weight of the scatterer in the sample. *I*(0)/*C*_W_ was plotted against the concentration of arrestin and analyzed based on the equilibrium model involving the equilibria between arrestin monomer and tetramer and the formation of the complex of arrestin and P-rRh.

### 4.5. Time-Resolved UV-Visible Spectroscopy

The binding of arrestin to P-rRh was measured by UV-visible spectroscopy using the shift of equilibrium between Meta-I and Meta-II towards the latter (extra-Meta-II). The mixture of P-rRh and arrestin was illuminated by a yellow flash, which was generated by a short arc xenon flash lamp and passed through a glass cutoff filter. The transient difference spectra after illumination were recorded using a time-resolved multichannel spectroscopy system with a time resolution of 100 µs (C10000 system, Hamamatsu Photonics, Hamamatsu, Japan) [45]. The temperature of the sample was kept at 20 °C. Spectral data were analyzed by IgorPro version 6.37 (WaveMetrics, Lake Oswego, OR, USA).

## Figures and Tables

**Figure 1 ijms-24-04963-f001:**
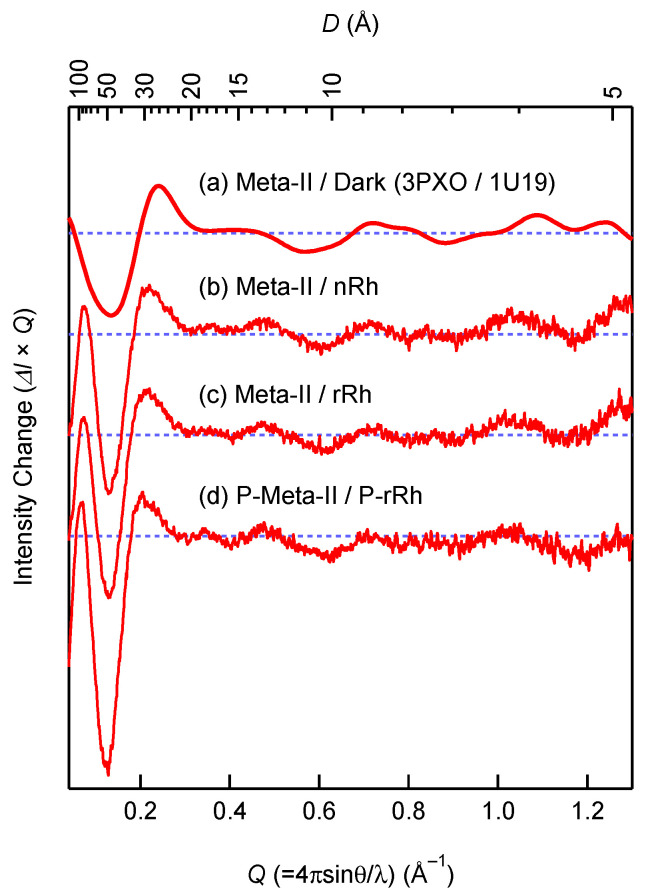
WAXS profile changes during Meta-II formation. (**a**) Calculated curve by CRYSOL [23], using the crystal structures of Meta-II (3PXO) [18] and the dark state (1U19) [24]. Diffrence WAXS curves between Meta-II and the dark state that are measured using (**b**) nRh/ND, (**c**) rRh/ND, and (**d**) P-rRh/ND.

**Figure 2 ijms-24-04963-f002:**
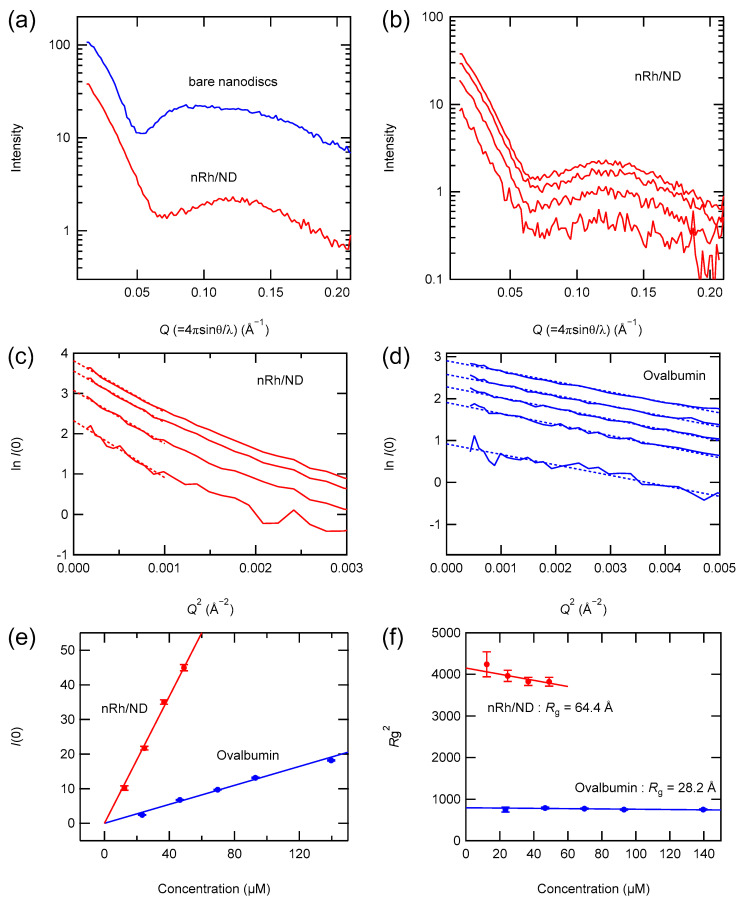
SAXS of nanodiscs containing nRh. (**a**) SAXS profiles of nRh/ND (red) and bare nanodiscs (blue). (**b**) SAXS profile of nRh/ND at 12, 25, 37, and 49 µM (from bottom to top, respectively). Guinier plot of (**c**) nRh/ND and (**d**) ovalbumin at 23, 47, 70, 93, and 140 µM (from bottom to top, respectively). The scattering curves in the small-angle region were linearly extrapolated to *Q*^2^ = 0, and *I*(0) and *R*_g_^2^ were estimated by the Y-intersections and slopes of regression lines, respectively (Equation (16)). (**e**) *I*(0) of nRh/ND and ovalbumin were plotted against the molar concentration (red and blue, respectively). The apparent molecular weight of nRh/ND was estimated to be 111 K from the slopes of regression lines and the molecular weight of ovalbumin (43 K) (Equation (17)). (**f**) *R*_g_^2^ of nRh/ND and ovalbumin were plotted against molar concentration (red and blue, respectively). *R*_g_ was estimated to be 64.4 Å for nRh/ND and 28.2 Å for ovalbumin from the Y-intersection of regression lines.

**Figure 3 ijms-24-04963-f003:**
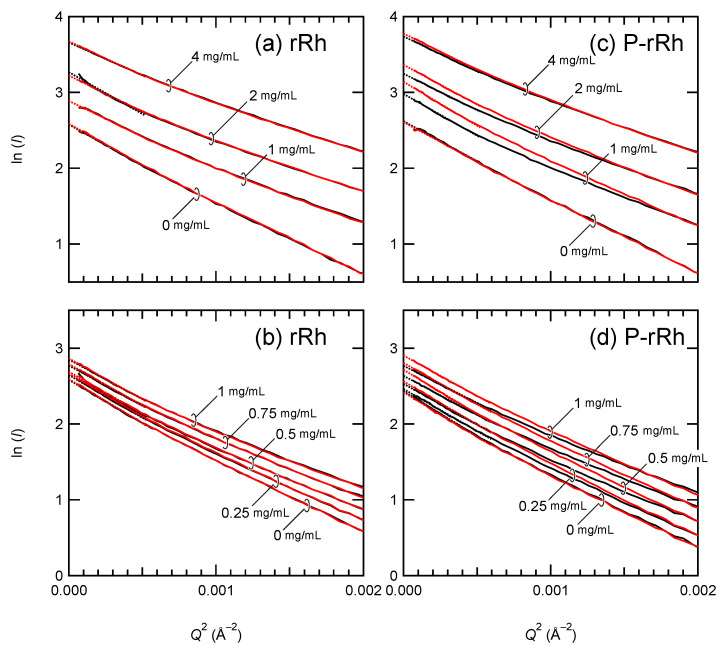
A Guinier plot of the mixture of arrestin and rRh/ND or P-rRh/ND. SAXS profiles were recorded before and after illumination (black and red, respectively). They were Guinier-plotted and linearly extrapolated to *Q*^2^ = 0 to estimate *I*(0). The concentrations of (**a**,**b**) rRh/ND and (**c**,**d**) P-rRh/ND were 20–23 µM, and those of arrestin (mg/mL) are shown in the figure.

**Figure 4 ijms-24-04963-f004:**
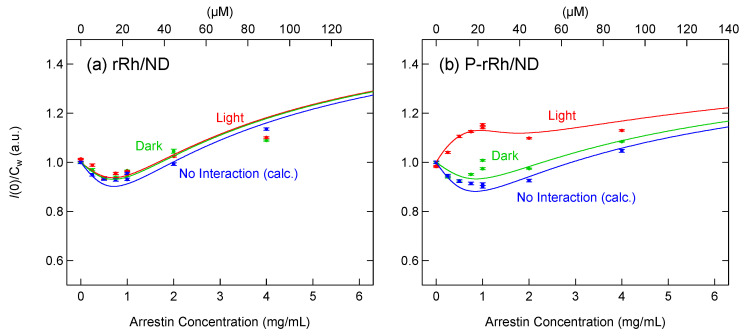
Light-induced formation of the complex of rhodopsin and arrestin. Various concentrations of arrestin were mixed with (**a**) rRh/ND or (**b**) P-rRh/ND (20–23 µM). *I*(0) before and after illumination (Figure 3) was divided by the weight concentration of total proteins (*C*_W_) and plotted against the total arrestin concentration (green and red, respectively). For comparison, the sum of *I*(0)/*C*_W_ of arrestin and (**a**) rRh/ND or (**b**) P-rRh/ND separately measured is also plotted (blue). It was fitted using the equilibration model to estimate $K_{\mathrm{complex}}$ (for details, see the text and Figure 5).

**Figure 5 ijms-24-04963-f005:**
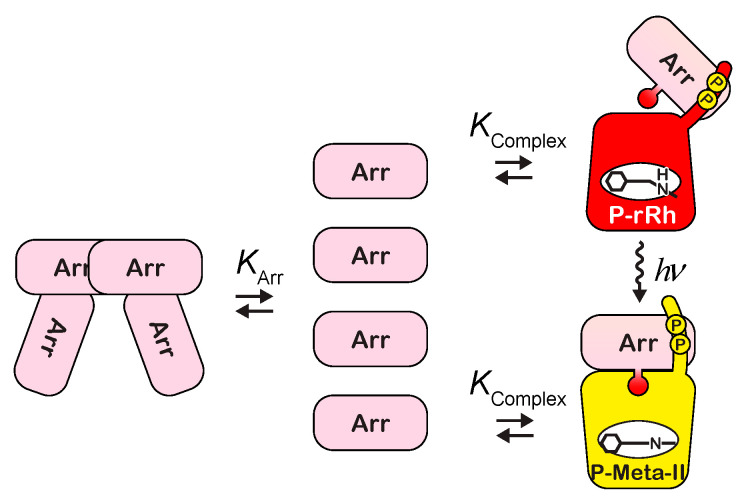
Interaction between phosphorylated rhodopsin and arrestin tested in the current experiments.

**Figure 6 ijms-24-04963-f006:**
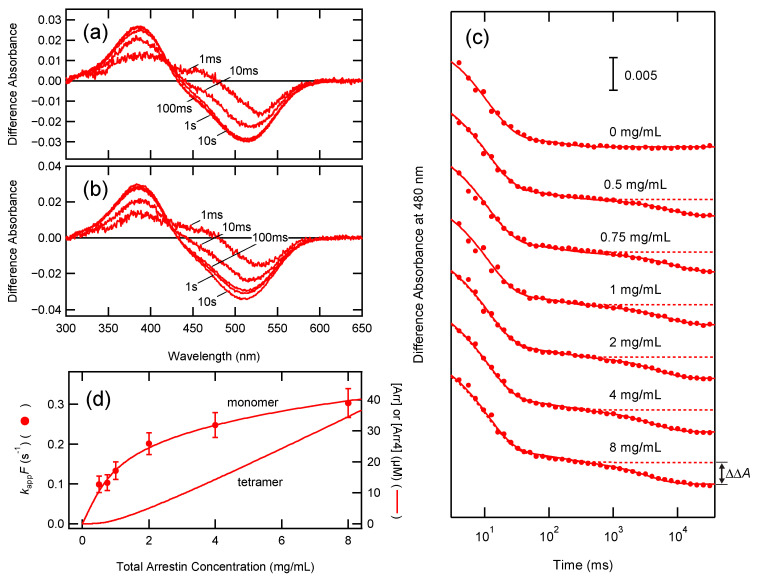
Extra-Meta-II formation induced by arrestin binding. Transient difference spectra after photoexcitation of rhodopsin (around 1 µM) in the (**a**) absence or (**b**) presence of 2 mg/mL arrestin were measured (transient difference spectra at 1 ms, 10 ms, 100 ms, 1 s, and 10 s are shown). (**c**) Changes in absorbance at 480 nm plotted against time after photoexcitation and fitted with a double- (0 mg/mL arrestin) or triple-exponential function (0.5–8 mg/mL arrestin), with the first and second components (transition from Meta-I to Meta-II) being global variables. For comparison, the fitting curve for 0 mg/mL arrestin is overlapped (dotted lines). The third component shows the formation of extra-Meta-II. The amplitude of the third component (ΔΔA) is proportional to the fraction of extra-Meta-II. (**d**) Rate constants for the formation of extra-Meta-II ($k_{\mathrm{app}}F=k_{a}\left[{\mathrm{Arr}}_{x}\right]$ ) are plotted against the arrestin concentration. The monomer and tetramer concentrations of arrestin based on Equation (1) are shown for comparison.

**Figure 7 ijms-24-04963-f007:**
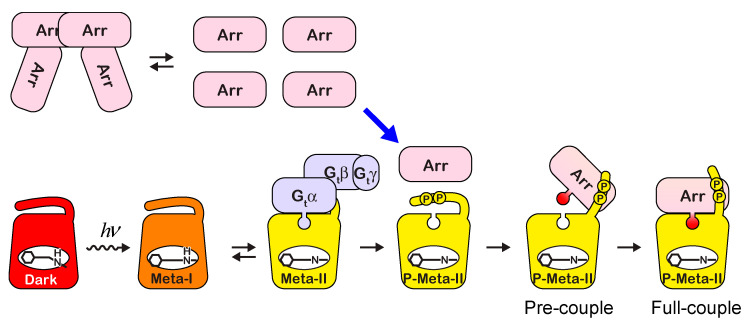
Monomer/tetramer equilibrium of arrestin and interaction with phosphorylated rhodopsin. Tetramer buffers monomer concentration to prevent fluctuations in quenching efficiency. Formation of the pre-coupling complex is essential to releasing the finger loop of arrestin.

**Table 1 ijms-24-04963-t001:** Dissociation constants of the arrestin tetramer and the rhodopsin/arrestin complex.

	$K_{\mathrm{Arr}}$ (µM) ^1^	$K_{\mathrm{complex}}$ (µM)
rRh/ND Dark	41.5	211
rRh/ND Light	41.5	167
P-rRh/ND Dark	41.5	109
P-rRh/ND Light	41.5	7.45

^1^$K_{\mathrm{Arr}}$ was estimated from Appendix A.

## Data Availability

Raw data are available upon request.

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
