# Peer review of "Role of Monomer/Tetramer Equilibrium of Rod Visual Arrestin in the Interaction with Phosphorylated Rhodopsin"

_ijms, 2023, doi:10.3390/ijms24054963_

Round 1

Reviewer 1 Report

Nanodiscs containing one rhodopsin molecule were prepared, and the binding process of monomeric and tetrameric arrestins to rhodopsin was investigated using the small-angle x-ray scattering (SAXS) and the wide-angle x-ray scattering (WAXS). It was found that the monomeric arrestin selectively binds to phosphorylated and activated rhodopsin. Interestingly, the tetramer of arrestin did not bind. This is a clear demonstration of the properties of arrestin by a physical method. The following points are comment.

1. In Figure 6D, the dependence of rate constant and monomer concentration on the concentration of arrestin is measured. The plotted monomer concentration does not match the rate constant. It appears to simulate by two straight lines instead of a curve. It would be better to explain why a curve instead of straight lines is used in Figure 6D.

2. Rhodopsin is phosphorylated by rhodopsin kinase when activated. Therefore, rhodopsin in the phosphorylated but not activated state may not be present in vivo. Explain the reason why experiment includes the phosphorylated but not activated state of rhodopsin.

Author Response

Thank you for your critical reading of our manuscript. Here are point-by-point responses to your comments.

Point 1: In Figure 6D, the dependence of rate constant and monomer concentration on the concentration of arrestin is measured. The plotted monomer concentration does not match the rate constant. It appears to simulate by two straight lines instead of a curve. It would be better to explain why a curve instead of straight lines is used in Figure 6D.

Response 1: In the original manuscript, the absorbance changes in Figure 6c were fitted by single- (0 mg/mL arrestin) or double-exponential functions (0.5-8 mg/mL arrestin) to show the trend of concentration dependence of the formation rate of extra-Meta-II. However, it appears to be oversimplified. In the revised version, we carried out more strict analysis in which association, dissociation, and equilibrium of arrestin/P-rRh complex were taken into consideration (section 2.4 and Eqs 7-14). The results demonstrated that the apparent rate constant of arrestin binding (kappF) well agreed with the concentration of monomer (Figure 6d), indicating that P-rRh specifically binds to arrestin monomer. These analysis was described in section 2.4 in detail and Figure 6 was totally revised. In addition, minor changes were made in Abstract, Introduction, and Discussion (section 3.1) as highlighted in the revised manuscript.

Point 2: Rhodopsin is phosphorylated by rhodopsin kinase when activated. Therefore, rhodopsin in the phosphorylated but not activated state may not be present in vivo. Explain the reason why experiment includes the phosphorylated but not activated state of rhodopsin.

Response 2: As you pointed out, rhodopsin is photoactivated and then phosphorylated in the cell. However, interaction between phosphorylated dark-state rhodopsin and arrestin is likely to mimic the pre-coupling complex. The following sentence was added to section 2.3.

"Because rhodopsin is phosphorylated after photoactivation in rod cells, this interaction would be an artifact, but mimics the pre-coupling, in which interaction between finger loop and cytoplasmic cavity is absent." 

Reviewer 2 Report

The authors studied interactions of arrestin with phosphorylated rhodopsin by the solution X-ray scattering method. They used the small-angle x-ray scattering (SAXS) to detect the protein-protein interaction as well as the wide angle x-ray scattering (WAXS) to detect the intramolecular rearrangement of the secondary structure elements. They carried out the WAXS and SAXS measurements using nanodiscs and tried to detect the photoactivation of rhodopsin, followed by the association with rod arrestin. They found that arrestin binds to phosphorylated rhodopsin in 1:1 stoichiometry even in the presence of significant concentration of tetrameric arrestin. Using the UV-visible spectroscopy, the authors found that the rate of the formation of rhodopsin/arrestin complex does not depend on the arrestin concentration.

In the Discussion the authors interpret the results obtained in terms of competition of two processes: interaction of monomeric arrestin with phosphorylated rhodopsin and formation of tetramers of arrestin. It would be useful to analyze this competition pattern more quantitatively in order to understand at what kinetic and equilibrium constants of these processes it can describe the experimental results. In the present version of the manuscript, the interpretation of the results looks too speculative.

Author Response

Thank you for your critical reading of our manuscript. Here is our response to your comment.

Point: In the Discussion the authors interpret the results obtained in terms of competition of two processes: interaction of monomeric arrestin with phosphorylated rhodopsin and formation of tetramers of arrestin. It would be useful to analyze this competition pattern more quantitatively in order to understand at what kinetic and equilibrium constants of these processes it can describe the experimental results. In the present version of the manuscript, the interpretation of the results looks too speculative.

Response: The time resolution of current SAXS experiments is about 5 sec, which is not enough to kinetic analysis of competition of two processes. However, in the equilibrium model of arrestin and phosphorylated rhodopsin to interpret SAXS data (Eqs. 1-5), both the tetramerization and complex formation are taken into consideration. Because original Figures 5b and 5c, in which tetramerization and complex formations were separately shown, appear to be misleading, these figures were revised to clearly show that KArr and Kcomplex are included in the simulation shown in Figure 4. In addition, it is noted again in Section 2.3 as highlighted in the revised manuscript.

Round 2

Reviewer 2 Report

I am satisfied by the revised version of the manuscript.